# Acetyl-CoA Synthetase 2 as a Therapeutic Target in Tumor Metabolism

**DOI:** 10.3390/cancers14122896

**Published:** 2022-06-12

**Authors:** Mengfang Liu, Na Liu, Jinlei Wang, Shengqiao Fu, Xu Wang, Deyu Chen

**Affiliations:** Institute of Oncology, Affiliated Hospital of Jiangsu University, 438 Jiefang Road, Zhenjiang 212001, China; liumf6@163.com (M.L.); liuna20190214@163.com (N.L.); wjl1990312@126.com (J.W.); fushengqiao@126.com (S.F.)

**Keywords:** ACSS2, tumor, metabolism, inhibitor, targeted therapy

## Abstract

**Simple Summary:**

Acetyl-CoA Synthetase 2 (ACSS2) is highly expressed in a variety of tumors, which is very important for tumor growth, proliferation, invasion, and metastasis in the nutritional stress microenvironment. Studies have proven that ACSS2 inhibitors can be effective in halting cancer growth and can be combined with other antineoplastic drugs to reduce drug resistance. This article mainly reviews the mechanism of ACSS2-promoting tumor growth from many aspects and the prospect of clinical application of targeted inhibitors.

**Abstract:**

Acetyl-CoA Synthetase 2 (ACSS2) belongs to a member of the acyl-CoA short-chain synthase family, which can convert acetate in the cytoplasm and nucleus into acetyl-CoA. It has been proven that ACSS2 is highly expressed in glioblastoma, breast cancer, liver cancer, prostate cancer, bladder cancer, renal cancer, and other tumors, and is closely related to tumor stage and the overall survival rate of patients. Accumulating studies show that hypoxia and a low serum level induce ACSS2 expression to help tumor cells cope with this nutrient-poor environment. The potential mechanisms are associated with the ability of ACSS2 to promote the synthesis of lipids in the cytoplasm, induce the acetylation of histones in the nucleus, and facilitate the expression of autophagy genes. Novel-specific inhibitors of ACSS2 are developed and confirmed to the effectiveness in pre-clinical tumor models. Targeting ACSS2 may provide novel approaches for tumor treatment. This review summarizes the biological function of ACSS2, its relation to survival and prognosis in different tumors, and how ACSS2 mediates different pathways to promote tumor metastasis, invasion, and drug resistance.

## 1. Introduction

Because of the fast metabolism of tumor cells, the oxygen demand is increased, and the energy supply is insufficient, so most tumor cells are in a state of stress. Studies have shown that tumor cells at a distance of about 100 μm from the blood vessels are in a state of hypoxia [1,2]. In theory, such a microenvironment is not conducive to tumor survival, but tumors can overcome this adverse environment and continue to maintain the ability of growth and proliferation. Because this environment will lead to changes in the epigenetics and proteomics of the tumor, the tumor will acquire a stronger ability to survive, invade, and metastasize, as well as become resistant to radiotherapy, chemotherapy, etc. [1,2,3]. It has been reported that acetyl-CoA is mainly derived from glucose and a small amount from glutamine and acetate under normal cell survival conditions. Under metabolic stress such as hypoxia or low serum, the raw material proportion of acetyl-CoA will be changed, and the proportion of acetyl-CoA produced by glucose will decrease, while acetyl-CoA from acetate will increase significantly [4,5]. The measured blood concentration of acetate is estimated to be about 20–50 μm, which is two orders of magnitude lower than the serum concentration of glucose [6]. How can such a low concentration of acetate provide the energy and raw materials required for the rapid growth and proliferation of tumor cells? However, isotope tracing experiments did observe a significant decrease in the utilization of glucose and a significant increase in the utilization of acetate, as well as a small increase in the utilization of glutamine under metabolic stress conditions [4,5].

ACSS2 is required for the uptake and utilization of acetate [7]. When ACSS2 was knocked out, the lipids and acetylated histones derived from acetate decreased significantly. ACSS2 can convert acetate to acetyl-CoA under nutritional stress. Acetyl-CoA is the central substance of the intracellular carbon source, which participates in the synthesis of many substances and acetyl group donors. ACSS2 is an enzyme located in the cytoplasm and nucleus, which is highly expressed in numerous tumors [8,9]. It participates in lipid synthesis in the cytoplasm and provides phospholipid membranes for rapid tumor proliferation [10,11]. In the nucleus, it can use local acetate to produce acetyl-CoA to provide acetyl groups for a variety of proteins [12] and can also activate its downstream molecules by acetylation. This review focuses on the role of ACSS2 in the cytoplasm and nucleus as well as the prospect of drug development for the targeted inhibition of ACSS2.

## 2. ACSS2

The currently known members of the mammalian acyl-CoA synthetase short-chain family are ACSS1, ACSS2, and ACSS3. ACSS1, mainly located in mitochondria, converts the substrate acetate into acetyl-CoA, which enters the tricarboxylic acid cycle to provide energy for cells [13,14,15]. The knockout of ACSS1 did not alter the level of histone acetylation but decreased the level of ATP [14,16]. ACSS3 is a hepatic mitochondrial matrix enzyme with a high affinity for propionate, and its expression level is up-regulated under starvation or fasting conditions [17]. There is also evidence that ACSS3 is necessary for acetate utilization and histone acetylation under metabolic stress and can also promote tumor growth and invasion [18,19]. Different research results may be caused by different cell lines and different physiological environments, which need to be further discussed.

ACSS2 is a well-studied member of the acyl-CoA synthase short-chain synthase family. ACSS2 expression is induced by sterol regulatory element binding protein 1 (SREBP1) [20]. Unlike other transcription factors, SREBP1 exists as a membrane-binding protein. Its active fragment can only enter the nucleus to activate transcription after it is released from the cell membrane [21]. But ACSS2 is not absolutely dependent on SREBP1 [22], because when SREBP is inhibited, the expression of ACSS2 is significantly decreased, but the basal level can be maintained [22]. SIRT1 (NAD+-dependent deacetylase1) regulates gene silencing, senescence, and energy metabolism; it deacetylates ACSS2 at lys-661 and activates it to play a role [23].

During the screening of functional genomes, ACSS2 was found to be a key gene for tumor cell growth during nutrition stress. Whether in a breast cancer cell line or a prostate cancer cell line, hypoxia and a low serum can significantly increase the expression of ACSS2, which in turn promotes cell survival under metabolic stress [5]. ACSS2 is a conserved enzyme existing in the cytoplasm and nucleus. It can be involved in the synthesis of various lipids in the cytoplasm [10,11], and can also translocate to the nucleus. In vitro experiments on human glioblastoma cell lines have shown that AMPK is activated by glucose deprivation or low serum levels. Activated AMPK phosphorylates ACSS2 at S659, which exposes the nuclear localization signal (NLS) of ACSS2, and the exposed NLS binds to the input protein α-5 on the nuclear membrane, which leads to the translocation of ACSS2 into the nucleus [12]. ACSS2 transferred to the nucleus also uses acetate as a substrate, and ACSS2 located in the nucleus is the only known enzyme that can utilize acetate [24]. Acetyl-CoA, derived from acetate, provides acetyl groups for various proteins in the nucleus, in particular, some key proteins, such as histone [16,25], hypoxia-inducible factor-2 alpha (HIF-2α) [26], transcription factor EB(TFEB) [12], and interferon regulatory factor 4(IRF4) [27]. ACSS2 in the nucleus can also promote the expression of tumor-related genes, thus promoting tumor growth, proliferation, invasion, and metastasis from various aspects. In addition, ACSS2 is tumor-specific [24], and its effect on tumors varies according to different tumor cell lines.

## 3. ACSS2 and Tumor Prognosis

### 3.1. Breast Cancer

Breast cancer is the second leading cause of cancer death in women after lung cancer [28]. Although the mortality rate is lower than before, the survival rate of advanced tumors is still low [29]. There are many treatments for breast cancer, but the fact of drug resistance and recurrence still cannot be changed. In vitro culture of breast cancer cell lines, hypoxia, and low serum can alter the lipid metabolism of breast cancer. ACSS2 was found to play a crucial role in increasing the proportion of lipids derived from acetate [5,6,25]. A high expression of ACSS2 was found in invasive ductal carcinoma, adenocarcinoma, and triple-negative breast cancer [7]. Moreover, ACSS2 was found to migrate to the nucleus under metabolic stress, and the amount of translocation was about three times higher than normal [25]. Triple-negative breast cancer is a special type of breast cancer with a poor prognosis [30]. Immunohistochemical analysis of tissue sections of human triple-negative breast cancer showed that patients with a high expression of ACSS2 had a lower overall survival, and there was a strong correlation between ACSS2 levels and disease progression (Table 1). The higher the tumor stage, the more significant the difference in ACSS2 expression between normal tissue and tumor tissue [5,25]. A clinical phenomenon has also been observed that breast cancer patients with a high expression of ACSS2 have a shorter survival time [5]. Moreover, immunohistochemical analysis of tumor tissues from 154 patients with breast cancer with a high expression of ACSS2 is associated with a shorter overall survival [7]. However, some studies have shown that breast cancers with a low ACSS2 expression have a worse prognosis [31] (Table 1).

### 3.2. Glioblastoma (GBM)

GBM is the most common type of malignant primary brain tumor, accounting for the majority of deaths among primary brain tumor patients, with an overall relative survival rate of only 6.8% at 5 years [40]. Although temozolomide was added to radiotherapy as an initial therapy in 2005, there has been no significant improvement in survival [41]. The unique brain microenvironment may drive different types of tumors to use the same substrate to promote tumor growth and invasion. Mice with brain metastases or glioblastoma can use acetate as a substitutable energy source [16]. The expression level of ACSS2 in glioblastoma is correlated with tumor grade, and the more ACSS2 that is expressed in cells, the higher the tumor grade (Table 1). The tumor cells expressing higher ACSS2 had a lower survival rate when the tumor was of the same grade [16] (Table 1). The injection of ACSS2shRNA glioblastoma cells into the ventral region of mice significantly inhibited tumor growth compared with the control group [12].

### 3.3. Hepatocellular Carcinoma (HCC)

The prevalence of liver cancer is gradually increasing, of which hepatocellular carcinomas account for approximately 90% [42]. Due to the high heterogeneity of hepatocellular carcinomas, the median survival of patients with cancer does not differ significantly, despite the availability of many therapeutic strategies [43,44]. Since hypoxia and energy deficiency are more common in the tumor microenvironment, simulating the tumor microenvironment by culturing hepatocellular carcinoma cells found that the expression of ACSS2 increased about five-fold compared to normoxia [7]. Immunohistochemical results showed that ACSS2 was mainly located in the tumor nucleus in HCC tissues with positive ACSS2 expression [34]. The loss of ACSS2 reduces the tumor burden of hepatocellular carcinomas in mice [7,10]; in mice with ACSS2-positive hepatocellular carcinoma, liver tumors grew faster and were more aggressive [7]. A new study showed that ACSS2 has two isoforms, ACSS2-S1 and ACSS2-S2, which are expressed in opposite ways in HCC and adjacent tissues. The progression-free survival time of patients with a high ACSS2-S2/S1 ratio is significantly shortened. The study clearly indicated that ACSS2-S2 promoted the growth and invasiveness of HCC cells in vitro [34].

### 3.4. Myeloma

Multiple myeloma is a malignant clonal disease of plasma cells [45]. Although the use of immune and targeted inhibitors and other therapies have led to a significant improvement in prognosis, with a median survival time of 10.5 years, it is still incurable as a precancerous lesion [46]. It has been shown that the expression of ACSS2 in myeloma-infiltrated plasmacytes is much higher than that in normal plasma cells, and that the knockdown of ACSS2 in myeloma with a high ACSS2 expression inhibits tumor cell proliferation and colony formation [27] (Table 1). A similar view was observed in vivo experiments [27]. In addition, obesity can promote the development of myeloma, but treatment with ACSS2i significantly reduced the incidence of obesity-induced myeloma from 60% to 20% [27]. Therefore, whether the application of ACSS2 inhibitors combined with other treatments for myeloma will improve the survival rate and reduce recurrence remains to be further studied.

### 3.5. Prostate Cancer and Bladder Cancer

Prostate cancer is the second most common cancer among men worldwide [47]. Metastatic prostate cancer is still clinically tricky. Although there are many prostate treatment methods, such as hormone therapy, chemotherapy, radiotherapy, immunotherapy, castration, and so on, the effect for advanced prostate cancer is still poor [48]. Many studies have found a high intake of acetate in prostate cancer [49,50]. It is well known that ACSS2 converts acetate to acetyl-CoA. Is ACSS2 also highly expressed in prostate cancer? Immunohistochemical observation showed that ACSS2 expression was high in metastatic prostate tumors, but relatively low in the surrounding normal tissues, and the expression level of ACSS2 in metastatic prostate tissues was higher than that in primary tumor tissues. The expression level of ACSS2 was correlated with the invasion degree of prostate cancer [5] (Table 1). Bladder cancer is also more common in men [47]. For bladder cancer, recurrence and drug resistance are the main reasons for the low individual survival rate [37]; the commonly used chemotherapeutic agent for bladder cancer is cisplatin. Studies have found that in patients receiving cisplatin chemotherapy and patients with complete remission of bladder cancer have lower levels of ACSS2, but patients with drug-resistant and progressive bladder cancer have higher levels of ACSS2 expression. Immunohistochemical (IHC) analysis of bladder tumors with different aggressiveness showed that the ACSS2 expression level was associated with the aggressiveness of bladder cancer [37,38] (Table 1). Further studies showed that the inhibition of ACSS2 could reduce resistance to cisplatin in bladder cancer [38].

## 4. ACSS2 Plays an Important Role in Tumor Promotion

### 4.1. ACSS2 Promotes Lipid Synthesis

About 50% of the key genes involved in tumor cell proliferation are related to lipid metabolism [51]. Lipids play an important role in almost all cellular activities [52,53]. Lipids are not only involved in constituting cell membranes [54,55,56,57], but also in cell-to-cell signal recognition, intracellular signal transduction, and energy supply [58,59,60,61], and even in promoting the immune escape of tumors [62,63]. Cancer cells increase de novo lipid synthesis [5,53,64], which in turn promotes their survival metastasis [64,65]. Acetyl-CoA for lipid synthesis is mainly derived from glucose under normal living conditions, but in vitro cultures of various tumor cell lines have demonstrated a significant increase in acetyl coenzyme A from acetate under metabolic stress [5,7,25,66]. ACSS2 in the cytoplasm converts captured acetate to acetyl-CoA, which is involved in the de novo synthesis of fatty acids mediated by fatty acid synthase (FASN) [10]. Moreover, ACSS2 can promote the expression of FASN [11] (Figure 1). In the human breast cancer cell line (BT474) a higher percentage of acetate was involved in lipid synthesis compared to normal metabolic conditions, which in turn could promote membrane phospholipid synthesis [5] (Figure 1). In vitro and in vivo experiments with four cell lines (lung cancer, breast cancer, melanoma, and colon cancer) demonstrated that hypoxia increased the expression of ACSS2 in tumor cells, which in turn increased the cellular uptake of acetate for lipid synthesis and promoted tumor progression [63] (Table 2). It has been shown that ACSS2 deficiency reduces fat deposition and obesity caused by a high-fat diet, but is less able to stress during fasting [10,67]. The researchers found that ACSS2-negative mice had a lower body weight and body fat content and a reduced body adipocyte volume than ACSS2-positive mice and that ACSS2-positive mice were more likely to develop a fatty liver by feeding the same litter of Acss2^+/+^ and Acss2^−/−^ mice with a high-fat diet. There was no significant difference in serum glucose, serum non-esterified fatty acid (NEFA), and ketone body levels between the two groups in the normal diet [10]. The knockout of ACSS2 has been shown to be normally viable in mice and has no effect on fertility or appearance, but ACSS2 is necessary for survival during fasting [67]. Although it has been shown that ACSS2 promotes neuronal differentiation and memory, knockout ACSS2 mice do not already cause severe behavioral changes compared to non-knockout mice and show similar levels in terms of exercise, coordination, body weight, and anxiety [68]. In vitro experiments on a variety of cancer cells have shown that ACSS2 expression determines the uptake and utilization of acetate [5,7,25]. Therefore, the tumor uptake of radioactive acetate can be used to observe the expression of ACSS2 in the cytoplasm and the lipid or lipid-soluble substances derived from acetate [69,70].

In addition, ACSS2 was also found to affect the LXR/RXR pathway (Figure 1 and Table 2). Forty-eight hours of fasting in knockout ACSS2 mice revealed a significant decrease in the expression of many genes downstream of LXR/RXR in the liver. LXR/RXR is a transcription factor that regulates lipid homeostasis during cholesterol synthesis, metabolic transport, and lipid synthesis [10,71,72]. These findings illustrate that ACSS2 can not only promote lipid synthesis in cells through acetate but also promote the expression of lipid-related genes to promote the expression of lipid metabolism genes (Table 2).

### 4.2. Role of ACSS2 in Autophagy

Autophagy is a highly conserved fine mechanism, which realizes the recycling of energy materials through the degradation of organelles and macromolecules [73]. Autophagy is bi-directional to tumor cells, and it can prevent the progression of cancer in the initial stage of the tumor. With the progression of the tumor, the metabolic pressure of the tumor microenvironment gradually increases, and autophagy can help tumor cells survive in this environment and promote their invasion and metastasis [31,74,75]. In the autophagy process of tumors, the role of ACSS2 gradually emerged. The high expression of ACSS2 in renal cell carcinoma promotes the expression of lysosome-associated membrane protein 1 (LAMP1) to promote tumor proliferation and invasion [36] (Figure 2). LAMP1 is an autophagy factor that not only promotes invasive cell metastasis but also assists with other factors to reduce the susceptibility of some factors to induce lysosomal cell death and increase the development of drug resistance [76,77,78]. The knockdown of ACSS2 reduces not only LAMP1 mRNA levels but also LAMP1 protein levels, which in turn leads to the invasion and metastasis of renal cell carcinomas [36]. In glioblastomas, metabolic stress can inhibit the mechanistic target of rapamycin complex 1 (mTORC1) and reduce TFEB phosphorylation, which in turn reduces the amount of TFEB bound to 14-3-3. The 14-3-3 proteins are a family of conserved regulatory molecules capable of binding to a variety of functionally diverse signaling proteins [79]. ACSS2, which is phosphorylated and translocated to the nucleus, promotes the entry of transcription factor EB (TFEB) into the nucleus and binds to TFEB [12] (Figure 2). TFEB promotes the expression of lysosome-related genes by promoting the expression of many key factors (CLEAR [80], RRAGC and UVRAG [81,82], CSTB [83], and M6PR and IGF2R [80]). Enhanced expression genes can enhance the function of lysosomes and promote autophagy, thus providing nutrition for tumor cells [80,84,85]. In addition, the ACSS2-mediated local acetate production of acetyl CoA provides acetyl groups for the acetylation of histone H3 around lysosomes and autophagy gene promoters [12] (Figure 2 and Table 2).

Recent studies have shown that ACSS2 promotes the expression of AGT5 by promoting the acetylation of H3K27 in the promoter region of autophagy-associated protein 5 (AGT5) [31], which promotes the survival of brain cells and neurons in the later stage of ischemia and hypoxia [86]. In a study related to autophagy in breast cancer, it was found that ACSS2 promoted AGT5 expression to enhance autophagy to inhibit the proliferation and metastasis of breast cancer cells induced by cadmium [31]. Another finding remains that the high expression of ACSS2 had a protective effect on breast cancer patients. It has been found that rapamycin binds to ACSS2 to promote the expression of ACSS2 to inhibit the progression of cadmium-mediated breast cancer [87]. These results are in stark contrast to many studies showing the role of highly expressed ACSS2 in breast cancer [5]. It is well known that ACSS2 has tumor specificity [24], and the opposite results of ACSS2-mediated autophagy in different tumors may be due to the complexity of autophagy, or the tumor specificity of ACSS2. However, the completely opposite effect in the same tumor raises the suspicion that ACSS2 has different isoforms and performs different functions in different cell lines of the same tumor. A recent study found ACSS2-S1 and ACSS2-S2 in hepatocellular carcinoma, and only ACSS2-S2 correlated with the malignancy of the tumor [34]. Whether there are other isoforms of ACSS2 to explain its functional diversity needs to be further explored.

### 4.3. ACSS2 Assists in Immune Escape

The tumor microenvironment facilitates tumor cells to escape from the attack of immune cells, i.e., immune escape of tumors, which is an important step in the occurrence and development of a tumor [88]. Regulatory T cells (Tregs) have been shown to play a key role in it [89,90]. The main role of regulatory T cells (Tregs) is to maintain physiological immune homeostasis by suppressing the effector T cells in different ways. Interferon regulatory factor 4 (IRF4) can enhance the inhibitory potential of Tregs and promote tumor growth in vivo [89,90]. ACSS2, which is highly expressed in patients with obesity-induced myeloma, converts acetic acid to acetyl-CoA, and lysine acetyl-transferase-binding protein (CBP) transfers the acetyl group from acetyl coenzyme A, which is mediated by the ACSS2 generation of acetic acid, to IRF4, resulting in the acetylation of IRF4 at K399. In this process, CBP-mediated acetyl group transfer is specific. Acetylated IRF4 reduces ubiquitinated degradation and increases protein levels to promote the differentiation of Tregs and the immunosuppression function of Tregs [27] (Figure 3). The ACSS2-IRF4 complex was found in myeloma using co-immunoprecipitation, and the knockdown of ACSS2 decreased the function of IRF4 and significantly inhibited the growth of myelomas [27,90] (Table 2). IRF4 is necessary for the survival of myeloma cells, and the deletion of IRF4 is lethal [91,92,93,94]. Inhibitors of ACSS2 can be used to accelerate its degradation and thus reduce the level of IRF4 and inhibit the growth of myelomas. In addition, some studies have found that ACSS2 can also promote tumor progression by affecting other immune cells and cytokines (Table 2). Immunohistochemistry of tumor tissues and adjacent normal tissues collected from 240 patients with cervical squamous cell carcinomas revealed that the expression of ACSS2 was significantly higher than that of adjacent normal tissues [39]. Moreover, nutritional stress will further increase the gap in ACSS2 expression between the two [95]. An analysis of the relationship between ACSS2 expression and immune cell-derived cytokines in cervical squamous cell carcinoma (CESC) using the TIMR database revealed that ACSS2 was significantly associated with the gene expression of immunosuppressive factors in CESC [39] (Figure 3). In cervical cancer, ACSS2 has also been shown to be associated with various immune cell infiltrates, especially with tumor-associated macrophages (TAM), which were not present in normal cervical tissue. TAM is a very important member in countering immune attacks and promoting tumor invasion and metastasis [96,97]. ACSS2 can also cooperate with Tregs to promote tumor metastasis and invasion [39]. Researchers also found that ACSS2 expression was positively correlated with PD-L1 expression but did not further explore whether ACSS2 could promote PD-L1 expression.

### 4.4. ACSS2 Fights Hypoxia

Hypoxia leads to changes in the metabolism of tumors, which in turn will increase the malignancy, treatment resistance, and metastatic ability of tumors, ultimately leading to a decrease in the survival rate of the tumor individual [1]. In this process, hypoxia inducible factor (HIF) plays an important role [2,98]. There are two isoforms of HIF-α, HIF-1α and HIF-2α, and the α-subunit of both isoforms is highly conserved at the protein level. Aerobic degradation and hypoxic accumulation mediated by tumor suppressor protein (VHL) after PHD hydroxylation are common pathways [99], but HIF-1α also has other specific pathways compared to HIF-2α [99,100]. Hypoxia stabilizes HIF-2α and promotes its entry into the nucleus, where ACSS2/CBP/SIRT1/HIF-2α signal transduction occurs, linking nutrient-sensing and stress signals to cell growth, invasion, and metastasis in mammalian tumors [26,27,98] (Figure 4). The expression of HIF-2α target gene in human fibrosarcoma cell line HT1080, a fast-growing cell line, was measured by semi-quantitative RTPCR under nutritional stress. It was found that the acetylation of HIF-2α in the presence of ACSS2 significantly increased the expression level of HIF-2α target gene and promoted the growth of tumor cell lines in vitro [26,101] (Table 2). When the basic amino acid in the nuclear localization signal of ACSS2 was changed to the acidic amino acid residue, phosphorylated ACSS2 was prevented from entering the nucleus, the level of HIF-2α acetylation was significantly reduced, and the HIF-2-dependent signal transduction was significantly weakened. It also inhibited the growth of human fibrosarcoma cells implanted in ventral mice and tumor cells cultured in vitro [26,101,102]. Amino acid alterations in nuclear localization signals in ACSS2 do not affect the expression of HIF-1α and HIF-1α target genes [1,26,101]; moreover, the formation and stabilization of the CBP/HIF-2α complex requires ACSS2 [101] (Figure 4). Researchers found that HIF is essential for normal development and that knocking out HIF-2α can be embryonic lethal [99,100]. Inhibitors have been developed, but they can cause drug resistance, anemia, leukopenia, and other adverse reactions [103]. Can the inhibition of ACSS2 be used as an adjunct to it to reduce the dose of drugs and consequently some adverse effects? Further studies are needed for ACSS2 inhibitors.

## 5. Prospect of Targeted Inhibition of ACSS2

ACSS2 expression is low in many normal tissues but has been found to be higher in many tumors than in the adjacent normal tissues. Its knockdown does not affect cell metabolism; it is essential for tumor cell growth under hypoxia and energy stress [5,7,10]. A number of studies have shown that ACSS2 gene deletion can inhibit the growth of a variety of cancers, including the above-mentioned breast cancer, prostate cancer, liver cancer, glioblastoma, esophageal cancer, kidney cancer, lung cancer, and other common tumors. It has been shown that (R)-1-ethyl-2-(hydroxydiphenylmethyl)-N-(2-hydroxypropyl)-1H-benzo(d)imidazole-6-carboxamide (also known as VY-3-135) is a potent ACSS2-targeting inhibitor, showing good specificity for ACSS2. Both in vitro and in vivo experiments in mice have demonstrated that VY-3-135 could inhibit tumor growth, and VY-3-135 tested in human breast models yielded similar results, namely, tumors with a high ACSS2 expression were more sensitive to treatment and significantly inhibited tumor growth [104]. Some research results provide answers to the question whether targeted inhibitors of ACSS2 can be used as adjuncts to reduce drug resistance. Recently, researchers have found that inhibition of ACSS2 can reduce drug resistance during treatment. For example, metastatic bladder cancer and esophageal cancer with a high expression of ACSS2 are more prone to developing cisplatin resistance, while the use of ACSS2 inhibitors in culture medium can reduce the drug resistance developed during cisplatin treatment [32,38]. In this paper, regarding ACSS2 mediating lipid metabolism, immune resistance, autophagy, and resistance to hypoxia to promote tumor growth, whether we can use ACSS2 inhibitors and in combination with some novel targeted inhibitors (such as lipid synthase inhibitor TVB-2640 [105]) to increase efficacy, and reducing the drug dosage and thus the drug side effects, need to be further studied.

Inhibitors of ACSS2 have been developed, such as novel substituted tetrazoles [106] and amide-substituted condensed pyridine derivatives [107] for the treatment of cancer. However, there are relatively few inhibitors of ACSS2. There are also some findings to the contrary: for example, rapamycin can effectively bind to ACSS2 to mitigate the cadmium-decreased levels of ACSS2 and thus inhibit the progression of breast cancer [31,87]. The opposite results in the same tumor, could it be that binding to rapamycin prevents it from working or decelerates its degradation, or again, that the selected cell line is not highly expressing ACSS2, looking forward to further exploration. Little is known about whether inhibitors of ACSS2 can reduce resistance to radiotherapy.

## 6. Conclusions

ACSS2 is highly expressed in a variety of tumors. It converts acetate into acetyl-CoA, which provides raw materials for tumor cell growth and metabolism and cellular activities. As a carbon intermediate, acetyl-CoA can participate in many biosyntheses and provides acetyl groups for various acetylations. Many key pathways in cells also require acetylation modifications that facilitate tumor adaptation to the nutrient-poor microenvironment. On the one hand, the knockdown of ACSS2 does not cause embryonic lethality in mice and basically does not affect the normal life of mice and has little effect on normal cells cultured in vitro. On the other hand, ACSS2 plays an important role for cell survival in many tumor cells. These provide good conditions for the development of targeted drugs. VY-3-135 has been developed, and its excellent effects and fewer side effects have been confirmed in vitro and in mice. However, it has only been tested in a breast cancer model, and its validation in other highly expressed tumors is lacking. In addition, ACSS2 inhibitors were found to reduce tumor resistance to cisplatin. However, there are inhibitors of ACSS2 combined with immunotherapy and radiotherapy to reduce tumor drug resistance and radiotherapy resistance, which is expected to be further studied.

ACSS2 is a new target for future tumor therapy and a tool for the assessment of its therapeutic effects. Currently, 11-C-acetate PET is a clinically mature imaging tool that has been widely used for cancer detection [108,109,110]. Imaging via 11-C-acetate PET can be easily used to help identify patients with tumors with a high uptake of acetate. It can also be used as a pharmacodynamic biomarker to longitudinally monitor the tumor response in patients treated with ACSS2 inhibitors [104]. ACSS2 inhibitors are a novel cancer treatment modality. However, some studies have shown that reducing ACSS2 can promote tumor progression and promoting ACSS2 expression can inhibit the growth of breast tumors [87]. Similarly, related studies in the gastrointestinal tract showed that reduced ACSS2 expression promoted tumor growth [111,112]. These different results need to be further explored. However, ACSS2 is a potential new target for tumors with high expression of ACSS2.

## Figures and Tables

**Figure 1 cancers-14-02896-f001:**
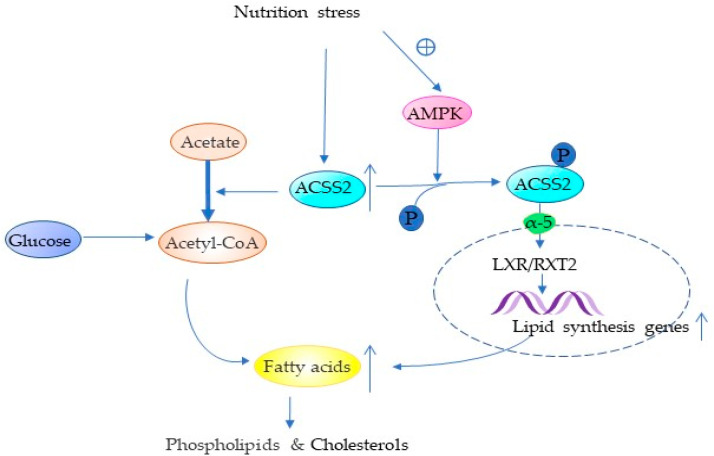
ACSS2 promotes lipid synthesis Nutrition stress resulted in AMPK-phosphorylation-dependent formation of ACSS2, which impacts the expression and activation of LXR/RXR to facilitate fatty acid and sterol metabolism. Moreover, ACSS2 increased the proportion of lipid synthesis derived from acetate in the cytoplasm. (Acetate and glucose line thickness indicate the proportion of participation.) α-5: importin 5. The arrows (↑) in the figures are used to highlight the increase in substance levels for easier understanding by the reader (The following figures also have the same meaning and will not be explained again).

**Figure 2 cancers-14-02896-f002:**
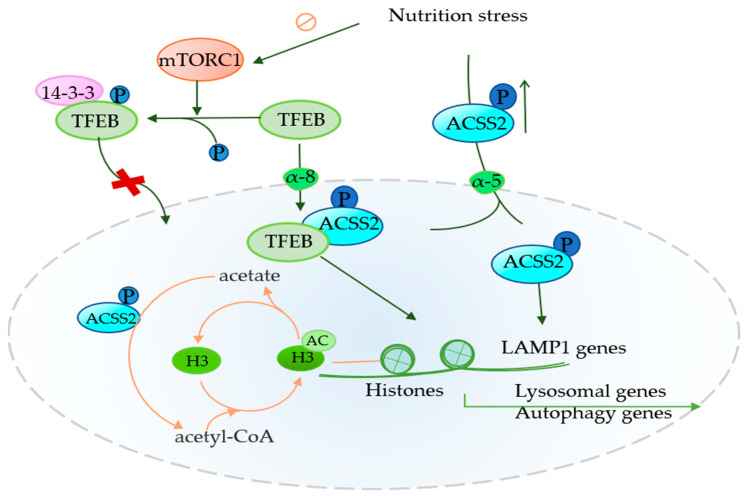
Role of ACSS2 in autophagy Phosphorylated ACSS2 enters the nucleus mediated by α-5 and forms a complex with TFEB mediated by α-8 in the promoter regions of lysosomal and autophagy genes, where ACSS2 incorporated acetate from the turnover of histone deacetylation into acetyl-CoA for histone H3 acetylation (the orange line segment represents the cycle between histone H3 and acetylated histone H3). Besides, ACSS2 enhance LAMP1 gene expression, which ultimately promote lysosomal biogenesis and autophagy. α-8: importin 8.

**Figure 3 cancers-14-02896-f003:**
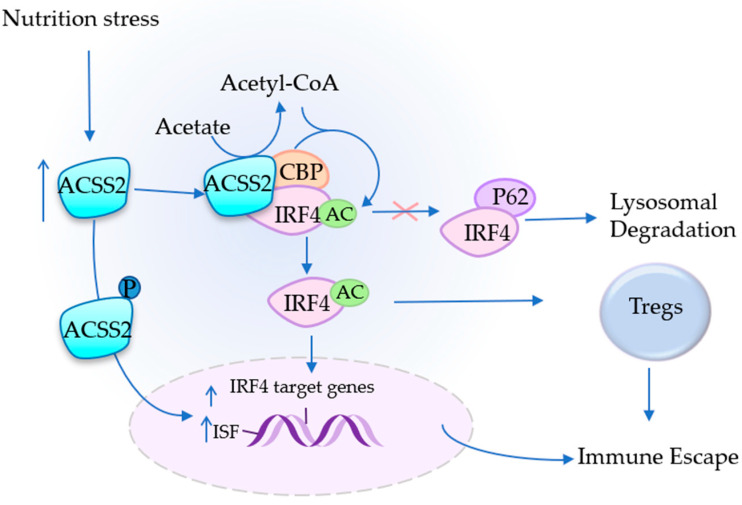
ACSS2 assists in immune escape ACSS2 not only participates in lipid synthesis but also acetylates IRF4 in the cytoplasm to prevent its degradation via ubiquitination (Ubiquitin binding protein P62 is a key molecule that mediates protein degradation.), but also Nucleus-translocatedACSS2 can also promote the gene expression of IRF4 and some immunosuppressive factors (ISF) and promote the immune escape of tumor cells.

**Figure 4 cancers-14-02896-f004:**
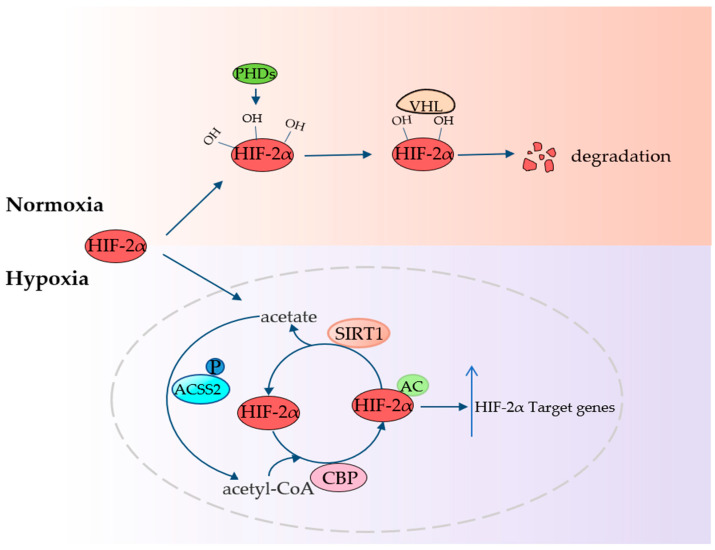
ACSS2 fights hypoxia HIF-2 α can be hydroxylated by PHD under normoxic condition, and then degraded by VHL. ACSS2 can use local acetate to produce acetyl-CoA, which provides the acetyl group for selective CBP acetylation of HIF-2α. Sirt-1 can deacetylate HIF-2α to produce acetate. The whole process continuously promotes the expression of HIF-2α target genes. PHD: prolyl hydroxylase.

**Table 1 cancers-14-02896-t001:** The relationship between ACSS2 and tumor invasiveness and patients’ survival time.

Systems	Tumors	ACSS2 Expression	Invasiveness	Survival	References
Nervous System	GBM	High	Up	Negative	[12,16]
Respiratory System	NSCLC	High	Up	Negative	[8]
Digestive System	EC	High	Up	Negative	[32]
	HCC	High	Up	Negative	[33,34]
		High	Down	Positive	[35]
Urinary System	RCC	High	Up	Negative	[36]
	Bladder Cancer	High	Up	Negative	[37,38]
Reproductive System	Breast Cancer	High	Up	Negative	[7]
		High	Down	Positive	[31]
	Prostate Cancer	High	Up	Negative	[5]
	Cervical Cancer	High	Up	Negative	[39]
Blood System	Myeloma	High	Up	Negative	[27]

Abbreviations: GBM, Glioblastoma; NSCLC, Non-small cell lung cancer; EC, Esophageal cancer; HCC, Hepatocellular carcinoma; RCC, Renal cell carcinoma.

**Table 2 cancers-14-02896-t002:** ACSS2 interacts with intracellular molecules and cells in different tumors.

Mechanisms and Metabolisms	Related Molecules and Cells	Effects	Cancers	References
Lipid metabolism	Acetate, FASN, LXR/RXR ways	Promotes lipid synthesis	NSCLC, BC, Melanoma	[5,7,25]
Autophagy	TFEB, Acetate, H3, LAMP1, TFEB	Promote autophagy	BC, GBM, RCC	[12,36]
Immunity	Acetate, IRF4, Tregs, TAMs, PD-L1	Assists in immune escape	MM, CESC	[27,39]
Hypoxia	Acetate, HIF-2α	Fight hypoxia	Fibrosarcoma	[26]

Abbreviations: BC: Breast Cancer; Many abbreviations have appeared in multiple contexts and will not be repeated.

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
