# Peer review of "Acetyl-CoA Synthetase 2 as a Therapeutic Target in Tumor Metabolism"

_cancers, 2022, doi:10.3390/cancers14122896_

Round 1

Reviewer 1 Report

 Through this review, the authors present the current state of research on the utility and problems of acetyl-CoA synthase 2 as a therapeutic target in tumor metabolism. The research area of molecular targeted drugs for cancer is of general interest, and furthermore, ASCC2 will be relevant to readers in broader areas such as epigenetic drug discovery.

The manuscript is generally well written and clear, and the references are logically presented. The manuscript provides an overview of gene product functions that is easy to understand for researchers who are not experts in this field, and the book is written in a compact manner to summarize the usefulness and problems of gene products as drug targets in relation to these functions.

The selection of references seems to be appropriate because it seems to be timely and up-to-date, and there is no bias toward one's own research or a particular study.

Figures and tables are kept to a minimum; a table summarizing the results of in vitro experiments could be added, which would help the reader to better understand this review.

Therefore, there are no major modifications to the text that I should suggest. If the following minor issues are corrected, this review is worthy of publication.

1) Table 1 is not relevant to the contents of Sections 3.3 and 3.4, so its mention in the text should be removed. 

2) The correspondence between the two columns of “systems” and “Tumors” in Table 1 is difficult to read and should be improved by correcting the line spacing.

3) In Figure 1, multiple pathways during cellular response are indicated by different colored lines. However, the background color of the illustration is not appropriate and is difficult to distinguish, so the background color should be changed. The explanation of each color line should be described in the figure legend.

Reviewer 2 Report

In this manuscript, the author reviewed the mechanism of ACSS2 in promoting tumor growth from different aspects and discussed the potential value of ACSS2 in clinical application. The manuscript is well organized and the key points are clearly stated. However, some minor points need to be taken into consideration when doing revision.

For the illustrator, the author included too much information in one figure, It feels crowded and obscures the critical point. I suggest the authors show the figures accordent to the subtitles.

For the language,  generally, the authors clearly express the idea, however, some sentences, grammar, and punctuations need to be corrected. 
